# Role of PITRM1 in Mitochondrial Dysfunction and Neurodegeneration

**DOI:** 10.3390/biomedicines9070833

**Published:** 2021-07-17

**Authors:** Dario Brunetti, Alessia Catania, Carlo Viscomi, Michela Deleidi, Laurence A. Bindoff, Daniele Ghezzi, Massimo Zeviani

**Affiliations:** 1Department of Medical Biotechnology and Translational Medicine, University of Milan, 20129 Milan, Italy; dario.brunetti@unimi.it; 2Unit of Medical Genetics and Neurogenetics, Fondazione IRCCS Istituto Neurologico Carlo Besta, 20126 Milan, Italy; alessia.catania@istituto-besta.it; 3Department of Biomedical Sciences, University of Padova, 35131 Padova, Italy; carlo.viscomi@unipd.it; 4German Center for Neurodegenerative Diseases (DZNE), 72076 Tübingen, Germany; michela.deleidi@dzne.de; 5Neuro-SysMed, Center of Excellence for Clinical Research in Neurological Diseases, Haukeland University Hospital, N-5021 Bergen, Norway; laurence.albert.bindoff@helse-bergen.no; 6Department of Clinical Medicine, University of Bergen, N-5021 Bergen, Norway; 7Department of Pathophysiology and Transplantation, University of Milan, 20122 Milan, Italy; 8Department of Neurosciences, University of Padova, 35128 Padova, Italy; 9Venetian Institute of Molecular Medicine, 35128 Padova, Italy

**Keywords:** PITRM1, pitrilysin metallopeptidase 1, mitochondrial proteostasis, Alzheimer Disease, neurodegeneration, neurodegenerative diseases, neurodegenerative dementia, spinocerebellar ataxia, mitochondrial protein quality control, protein aggregation, mitochondrial dysfunction

## Abstract

Mounting evidence shows a link between mitochondrial dysfunction and neurodegenerative disorders, including Alzheimer Disease. Increased oxidative stress, defective mitodynamics, and impaired oxidative phosphorylation leading to decreased ATP production, can determine synaptic dysfunction, apoptosis, and neurodegeneration. Furthermore, mitochondrial proteostasis and the protease-mediated quality control system, carrying out degradation of potentially toxic peptides and misfolded or damaged proteins inside mitochondria, are emerging as potential pathogenetic mechanisms. The enzyme pitrilysin metallopeptidase 1 (PITRM1) is a key player in these processes; it is responsible for degrading mitochondrial targeting sequences that are cleaved off from the imported precursor proteins and for digesting a mitochondrial fraction of amyloid beta (Aβ). In this review, we present current evidence obtained from patients with *PITRM1* mutations, as well as the different cellular and animal models of *PITRM1* deficiency, which points toward PITRM1 as a possible driving factor of several neurodegenerative conditions. Finally, we point out the prospect of new diagnostic and therapeutic approaches.

## 1. Introduction

Mitochondrial dysfunction, whether primary or secondary, is increasingly recognized as a hallmark of neurodegeneration [1] and a wide body of literature provides evidence of impaired mitochondrial function as a cause rather than a consequence of neurodegeneration [2,3,4]. Mitochondrial demise can be observed even before the appearance of pathognomonic histopathological hallmarks of the disease [5]. Neurons are obligatorily dependent on mitochondrial energy production, which sustains a myriad of functions, including membrane remodeling, synaptic spine formation, and the generation of transmembrane resting and action potentials [6,7].

Mitochondria also play a pivotal role in cell survival and death by regulating apoptotic pathways and contributing to different cellular functions including intracellular calcium homeostasis, maintenance of the cellular redox potential, and cell cycle regulation. Recent evidence demonstrated that mitochondria are also important in regulating cell fate towards stemness or neurogenesis [8].

Thus, it is not surprising that mitochondrial dysfunction can have devastating effects on neuronal differentiation and survival. The brain is a major target in primary, genetically determined mitochondrial disease, but mitochondrial dysfunction is also a prominent feature in many of the most prevalent neurodegenerative diseases, including Parkinson’s Disease (PD) [9], Huntington’s Disease [10], neurodegeneration associated with stroke [11], Amyotrophic Lateral Sclerosis [12], neurodegenerative ataxias [13] and different types of psychiatric and cognitive disorders such as Dementia with Lewy Bodies [14], Frontotemporal Dementia [15], and Alzheimer’s disease (AD) [16]. In AD, for example, the “mitochondrial cascade” hypothesis proposes that organellar dysfunction is the primary event in AD pathology [17]. Moreover, although the extracellular deposition of amyloid-beta (Aβ) peptides (Aβ_1–40_, Aβ_1–42_) is the key histopathological hallmark of AD, Aβ accumulation is proposed to also occur in mitochondria, through the mitochondrial import machinery [18,19], causing impairment of different mitochondrial pathways such as respiration, reactive oxygen species (ROS) detoxification, and organelle morphology [20,21,22,23,24,25,26].

The protease-mediated quality control system is a first-line homeostatic defense against mitochondrial damage, and includes degradation of non-assembled, misfolded or damaged proteins as a result of oxidative stress, elimination of cleaved products during protein processing, and overall regulation of protein turnover and homeostasis (referred to as proteostasis) [27]. The proteolytic system in mitochondria is crucial for the maintenance of protein turnover and the integrity of mitochondria. The majority of mitochondrial proteins are synthesized on cytosolic ribosomes with an N-terminal peptide (the presequence or mitochondrial targeting sequence—MTS) which is recognized by—and binds to—specific receptors in the mitochondrial outer membrane. Following this event, these precursor mitochondrial proteins are translocated through the mitochondrial entry gate, the TOM (Translocase of Outer Membrane) complex and then, via a specific TIM (Translocase of Inner Membrane) system, TIM23, into the matrix [28], where they undergo proteolytic processes, including the cleavage of MTS, and structural modifications that lead to mature, functional proteins [29].

Several proteases and peptidases have been identified in different mitochondrial sub compartments. Most are ATP dependent proteases, such as the matrix-located Lon protease 1 (LONP1) and the membrane-bound AAA (ATPases Associated with diverse cellular Activities) proteases. The latter include i-AAA (Yme1), active in the intermembrane space (IMS), and m-AAA (Yta10/Yta12), exposed to the matrix. These enzymes catalyse the initial step of degradation, cleaving proteins into peptides, thus contributing to mitochondrial quality control [30]. Other ATP-independent proteases such as the mitochondrial processing peptidase (MPP), the mitochondrial intermediate protease (MIP), and the inner membrane peptidase (IMP), also generate short peptides. Upon import of the mitochondrial precursor proteins, MPP in the matrix cleaves the MTSs, releasing the mature proteins [31,32,33]. MPP activity can produce import intermediates, which are further processed by MIP (the octapeptidyl peptidase Oct1 in yeast) or IMP (the intermediate cleaving peptidase Icp55 in yeast) [33,34,35,36]. Therefore, all these proteases are involved in the processing of precursor protein and release free MTS peptides after proteolytic cleavage (Figure 1) [37].

Typically, the MTSs are amphiphilic species, with a polar, positively charged, arginine-rich side, opposite to an apolar side (Figure 1). Therefore, if these MTS peptides fail to be cleared from the mitochondrial matrix, they may act as detergent-like, toxic agents, forming pores in the membranes and resulting in uncoupling of oxidative phosphorylation and dissipation of the mitochondrial membrane potential [36].

## 2. PITRM1/PreP

PITRM1, also termed Presequence protease (PreP), is localized in the mitochondrial matrix; it is the only known protein responsible for the degradation of the MTS, thus completing the last step of the protein import process. PreP was initially identified in *Arabidopsis thaliana* (AtPreP), and shown to degrade the MTS of both mitochondria and chloroplasts [38]. A yeast mutant lacking the *PITRM1* homolog *Cym1* [39] displayed mitochondrial accumulation of precursor proteins and processing intermediates, as well as decreased levels of cleaved, mature proteins [32,40]. Impaired preprotein maturation leads to accelerated protein degradation and an unbalanced mitochondrial proteome, resulting in mitochondrial dysfunction manifested by reduced respiration, altered membrane potential, and high ROS levels [28,37].

The human gene *PITRM1* is located in chromosomal region 10p15.2, contains 27 exons, and is transcribed from the antisense strand. Three different isoforms of the protein correspond to transcript variants derived from alternative splicing. In isoforms 1 and 2, all 27 exons of the gene are retained and translated, with a total of 1038 and 1037 aminoacids (aa), respectively, due to the use of two different splice donor sites at exon 17. The third isoform is the smallest one and contains 939 aa derived from 24 exons of the gene.

Human PITRM1 was initially identified as metalloprotease 1 [41], and shows 31% sequence identity to AtPreP, performing a similar function in human mitochondria. PITRM1 belongs to the M16 metalloproteases family, which are Zn^2+^-dependent and ATP-independent enzymes that share a conserved architecture of two ~ 50 kDa homologous domains enclosing a large catalytic chamber. Three families, M16-A, -B and -C, have been characterized depending on the connection between these two domains [42]. PITRM1 is a 117 kDa M16C enzyme arranged in four domains, forming two enzyme halves, hPreP-N (amino acids 33–509) and hPreP-C (aa 576–1037) domains, connected by a hinge region (aa 510–575), which is possibly involved in the opening and closing of the two enzyme halves. Active site residues are located in both halves, which interact to form a large peptidasome chamber [43]. In addition to targeting peptides generated during the import processing, the large catalytic chamber of PITRM1 is also able to degrade a wide range of unstructured peptides, ranging from 10 to 65 aa, but not larger folded proteins [44].

In vitro studies demonstrated that human PITRM1 can degrade Aβ_1–40_, Aβ_1–42_ and the Aβ arctic (Aβ_1–40_ E22G), a peptide that causes increased fibril formation and early onset of a familial variant of AD. Its proteolytic activity generates several fragments, which are unique to PITRM1 (in comparison to other proteases), and recognizes the cleavage sites in the very hydrophobic C-terminal portion of Aβ that is prone to aggregation [38]. Interestingly, the 3D structures of PITRM1 are highly similar to IDE (Insulin degrading enzyme), a zinc metallopeptidase that degrades intracellular insulin, as well as glucagon, amylin, beta-amyloid, bradykinin, and kallidin. Moreover, IDE has been suggested to have a role in the degradation of cleaved MTSs [37,45]. The preferential affinity of this enzyme for insulin results in the insulin-mediated inhibition of degradation of other peptides, such as beta-amyloid [46]. Variants in the *IDE* gene and deficiencies in the protein function have previously been associated with type 2 diabetes mellitus and Alzheimer’s disease [47,48].

## 3. PITRM1/PreP and Alzheimer’s Disease

### 3.1. PITRM1/PreP Is Downregulated in AD Patients and Mouse Models

It was reported that mitochondria isolated from the temporal lobe, an area of the brain that is highly susceptible to Aβ accumulation, showed significantly lower PITRM1 activity in AD patients compared to age-matched control samples. In contrast, activity in mitochondria isolated from the cerebellum showed no differences [49]. Decreased expression of a transcript antisense to *PITRM1* was found in the posterior cingulate cortex extracted from AD patients, suggesting changes in the regulation of *PITRM1* expression [50].

Further, global proteomic studies revealed a mitochondrial proteome imbalance in AD samples, with presequence-containing mitochondrial proteins being particularly affected [23,51]. Similar experiments were performed in transgenic (Tg) mouse models of AD, including a Tg for *APP* (Amyloid-β Protein Precursor), which is the most studied AD model, and ABAD (Aβ-binding alcohol dehydrogenase), which is an accelerated AD model. Indeed, Aβ peptides are produced by regulated proteolysis of the APP through the action of β- and γ- secretases. ABAD or HSD17B10 interacts with Aβ and mediates its mitochondrial toxicity [20]. The proteolytic activity of mouse PreP (mPreP) was significantly reduced in the matrix of cortical mitochondria isolated from 5-month-old Tg mAPP mice compared to non-Tg littermates. Similarly, the mPreP activity in Tg mAPP/ABAD mice was significantly lower than that in both non-Tg mice and Tg mAβPP mice [49]. Transgenic mice overexpressing *APP* also showed lower mPreP proteolytic activity compared to age-matched, non-transgenic mice and a progressive, age dependent reduction in the same activity [49].

These results suggest that the loss of PITRM1 activity may result not only in the accumulation of Aβ, but also of free presequence peptides [36]. While the relevance of Aβ accumulation still remains to be determined, the accumulation of presequence peptides is likely to be toxic.

The lower PITRM1 activity observed in AD brain mitochondria appears not to be due to lowered protein levels but to a functional alteration of the enzyme, possibly through post-translational modifications such as protein oxidation. Oxidative stress is well-documented in AD brains and confirmed by the increase in several reactive oxygen species (ROS) markers [52]. For example, levels of 4-hydroxynonenal derived from lipid peroxidation, a biomarker of oxidation, were found to be higher in the temporal lobe of AD patients than controls, while were unchanged in the cerebellum [49]. These results suggest that lower temporal lobe PITRM1 activity reflects protein oxidation, and this is confirmed by several biochemical analyses using purified PITRM1 [53]. In addition, it has been shown that PITRM1 can be inactivated by the oxidative-dependent formation of disulfide bridges [43], which can be reverted by the antioxidant enzyme methionine sulfoxide reductase A (MsrA) [53].

Interestingly, it appears that PITRM1 activity is not only important for degrading mitochondrial Aβ; it also appears to influence total brain Aβ levels, suggesting that mitochondrial Aβ somehow plays an important overall regulatory role. Since exogenous or intra-cellular Aβ can enter mitochondria via the aforementioned pathways, the mitochondrial pool of Aβ may undergo dynamic changes, contributing to the balance of intra-cellular/extra-cellular Aβ accumulation.

Increased neuronal mPreP in double transgenic *mAPP/PreP* mice not only reduced Aβ accumulation in the brain but also remarkably suppressed the expression of the Receptor for Advanced Glycation End-products (RAGE) as compared with mutant *APP* mice. RAGE are important cell-surface receptor-mediating chemotactic and inflammatory reactions that occur in response to Aβ and other proinflammatory ligands [54,55]. RAGE signalling is known to promote the activation of microglia (as shown by increased expression of microglial markers e.g., CD4 and CD11) in the Aβ-enriched brain, and the production of proinflammatory mediators including cytokines and chemokines. These results suggest the involvement of RAGE in PreP/Aβ-mediated cytokine and chemokine production along with microglial activation [56].

### 3.2. Targeting PITRM1/PreP as a New Therapeutic Strategy for Treatment of AD

Besides AD patients, reduced levels of mPreP were found in both the hippocampus and cortex of the APPswe/PS1dE9 (APP/PS1) transgenic mouse, an AD model that develops spatial memory impairment, increased Aβ deposition in the brain, synaptic loss, and mitochondrial dysfunction, features that are similar to those observed in AD patients. Importantly, 12 weeks of treatment with Ligustilide (LIG, 3-butylidene-4, 5-dihydrophthalide) in 7 month old APP/PS1 mice increased PreP protein levels and reduced mitochondrial Aβ_1–42_ concentration in both the hippocampus and neocortex [57]. Recently, an age-dependent decrease in mPreP levels was reported by our group [58] in the hippocampus of the Senescence Accelerated Mouse Prone 8 (SAMP8), a model characterized by high oxidative stress and cognitive decline, with the onset of an AD–like phenotype between 9 and 15 months of age. Stimulation of the Sirt1-PGC-1α axis by a metabolic activator increased mPreP protein levels and improved the cognitive dysfunction [59].

A potential role for PITRM1 in AD is also fostered by studies showing that increased expression and activity of neuronal mPreP significantly reduced the mitochondrial Aβ load, and improved mitochondrial function, and synaptic plasticity and strength, in AD mouse models, and prevented the development of impaired spatial learning and memory. This enhancement of PITRM1 activity provided a new therapeutic option for the treatment of AD [56,60]. Accordingly, there has been an increasing interest in the resolution of PITRM1 structure to develop specific agonists; benzimidazole derivatives (3c and 4c) were reported to act as agonists that boost PITRM1 proteolytic activity [61]. Unfortunately, pharmacological attempts using this approach have, so far, been unsuccessful [62] and therapeutic modulators of PITRM1 remain to be identified.

## 4. Genetic Variants in Human *PITRM1*

Perturbation of PITRM1 activity appears to influence Aβ accumulation, making *PITRM1* a candidate for diseases such as AD in which amyloid accumulates [43]. Nevertheless, a conclusive association between *PITRM1* variants and AD has not yet been demonstrated [63]. In a genome-wide association study using microarray technology, 18 single nucleotide polymorphisms (SNPs) were found and genotyped in 673 AD cases and 649 controls in a Swedish population. AD cases comprised 256 unrelated familial AD subjects, having at least one affected first- or second-degree relative but no mutation detected in the *APP, PS1 or PS2* genes, and 417 late onset clinical AD cases without a positive family history. No significant difference in genotype distribution between cases and controls, and no correlation between these mutations and AD progression could be identified. The SNPs analysed included nine tagged SNPs selected from the CEPH European panel in the Hapmap database with a minor allele frequency >0.20, and eight additional coding SNPs located in exon 4, which encodes the PITRM1 active site. However, a functional analysis of several PITRM1 SNP variants, selected on the basis of localization of the substituted amino acid in the enzyme structure, showed decreased activity in comparison to wild type PITRM1 [64].

### Biallelic Mutations in PITRM1 Are Associated with Complex Phenotypes and Amyloidotic Neurodegeneration

We recently found that recessive *PITRM1* mutations are associated with a slowly progressive syndrome characterized by mental retardation, spinocerebellar ataxia, cognitive decline and psychosis [65]. The first family described comes from a small Norwegian coastal community comprising <200 individuals. Out of five siblings, two were unquestionably affected, one additional sibling had peripheral neuropathy, another had psychiatric symptoms but refused investigation, and one died of cancer before DNA collection.

The index case (II-2) was diagnosed with mild intellectual disability as a child and later developed progressive spinocerebellar ataxia, obsessional behaviour and psychotic episodes with hallucinations. Brain MRI showed marked cerebellar atrophy and unilateral signal changes in the thalamus. The routine blood profile was unremarkable, but cerebrospinal fluid (CSF) examination showed low levels of Aβ_1–42_ (363 ng/L; n.v. > 550), similar to the ones detected in idiopathic AD [66,67]. Total and phosphorylated Tau and 14-3-3 proteins were normal. A muscle biopsy showed some scattered COX-negative fibres. Respiratory chain (RC) complex assays in muscle homogenate from individual II-2 showed low specific activities of all RC complexes, with a concomitant decrease in citrate synthase (CS), an index of mitochondrial mass. Her brother (II-4) was reported to be mildly mentally retarded from an early age, and also developed obsessional behaviour with episodes of psychosis, and early onset ataxia. His CT scan showed cerebellar and milder cerebral atrophy.

SNP-based homozygosity mapping and whole exome sequencing (WES) revealed a c.548G > A p.R183Q mutation in *PITRM1*. The *PITRM1^R183Q^* mutation resulted in a marked reduction in *PITRM1* protein levels in patients’ fibroblasts and skeletal muscle cells. In vitro, its catalytic activity was comparable to wild-type protein. However, when modelled in yeast, the R183Q substitution resulted in lower mitochondrial oxygen consumption and cytochrome content and reduced capacity to degrade Aβ. A recent study [68] showed that the structural and electrostatic properties of the conserved strand-loop-strand motif containing *PITRM1* residue R183 are critically important for *PITRM1* function. The mutational disruption of electrostatic interactions in proximity of *PITRM1* residue R183 contributes to the loss of enzyme activity and may contribute to the loss-of-function phenotype observed in *PITRM1* R183Q-dependent neuropathy. However, PITRM1 activity was not tested directly, but deduced in vitro using the recombinant protein and fluorogenic substrates rather than native presequence peptides.

A novel *PITRM1* mutation causing autosomal recessive spinocerebellar ataxias (ARCA) in four children from two independent families was recently identified in Jerusalem, Israel [69]. Affected patients from both families were born to consanguineous parents and the onset of the disease was in early childhood. Even though they carried the same *PITRM1* variant (PITRM1T931M) on a shared genetic haplotype, they showed a significantly different phenotype. This is likely attributable to an additional chromosome X rearrangement found in the first family; indeed, homozygous brothers belonging to this pedigree presented with a more severe disease course characterized by delayed psychomotor development, severe ataxia and psychotic episodes, somehow resembling the Norwegian family above mentioned. Brain MRI revealed cerebellar atrophy, while signs of muscle involvement (elevated plasma CPK and lactate/pyruvate) were detected in one of them. A milder psychomotor retardation was reported in both siblings from the second Palestinian family, with one subject also presenting with cerebellar signs. Both patients from this second family had normal MRI and routine blood tests.

Again, functional studies performed on patients’ fibroblasts transduced with the recombinant wild-type protein, and, in a yeast model, demonstrated that this mutation leads to loss of function shown by a defective cleavage of >40 amino acid peptides, including Aβ species, strengthening the connection between impairment of mitochondrial peptide degradation and neurodegenerative diseases [69].

A very recent case report described a new patient harboring a homozygous variant, c.2239dupG, in *PITRM1* due to segmental maternal isodisomy of chromosome 10. The patient presented at 25 months of age with seizures, hypotonia and delayed motor and language development. At 36 months of age, the patient evidenced a severe ataxic syndrome associated with intellectual disability and persistence of epilepsy. Unlike previously reported cases, this patient did not show any psychiatric symptoms until the last follow up examination (at 6 years). Besides a moderate cerebellar atrophy, brain MRI also revealed bilateral thalamic hyperintensity.

The patient’s fibroblasts had severely reduced *PITRM1* transcript and protein expression, and the measurement of oxygen consumption rates denoted impaired mitochondrial ATP metabolism due to defective OXPHOS [70].

Thus, robust evidence shows that PITRM1 deficiency due to biallelic variants leads to progressive neurological impairment including ataxia, mental retardation and psychiatric features, and an overall susceptibility to neurodegeneration (Table 1).

## 5. Animal Models with Genetic *PITRM1* Impairment

### 5.1. Mouse Models

A *Pitrm1* knock-out C57BL/6n-Atm1Brd mouse model was developed by the Wellcome Trust Sanger Institute, Cambridge, UK, and first described by our group [65]. Whilst the constitutive *Pitrm1*^−/−^ genotype is associated with embryonic lethality, *Pitrm1*^+/−^ heterozygotes survived to adulthood and reduced the levels of Pitrm1 protein in different tissues (50% in brain and liver, and approximately 60% in skeletal muscle), thus replicating the haploinsufficiency-like effect of PITRM1^R183Q^ patients (reduced amount of a catalytically normal enzyme).

*Pitrm1*^+/−^ mice present with an early onset, slowly progressive neurological phenotype starting with a clasping reflex and evolving into an ataxic phenotype with reduced performance in coordination tests. Metabolic assessment showed significantly reduced O_2_ consumption and CO_2_ production, and reduced heat production. Western blot immune analysis on 4-month old (mo) brain homogenates revealed an approximately 2.5-fold increase in APP cross-reacting material in *Pitrm1*^+/−^ vs. *Pitrm1*^+/+^ specimens. Neuropathological analysis of 4–12 mo mice revealed increased gliosis, accumulation of ubiquitin-positive material in the neuropil and neurons, and increased reactivity to APP and Aβ_1–42_ antibodies. Furthermore, scattered Thioflavin and Congo red-positive areas were found in the brain cortex, indicating the presence of amyloid deposits. The import of Aβ_1–42_ into isolated mitochondria was studied in *Pitrm1*^+/+^ and *Pitrm1*^+/−^ mice and confirmed a reduced capacity of *Pitrm1*^+/−^ brain and liver mitochondria to degrade Aβ_1–42_, causing this peptide to accumulate. Using in-vitro import experiments, we also demonstrated an accumulation of MTS in *Pitrm1*^+/−^ mitochondria.

In a previously published paper [65], we described the clinical phenotype observed in the mouse during the first 12 months of age. The evolution of symptoms for up to 24 months (unpublished data) included hind limb paresis and tremors at the age of 15 months, and significant cognitive deficit at the Y maze after 12 months of age (unpublished data). These findings suggest that not only recessive homozygous mutations (as in the *PITRM1* families), but also haploinsufficiency associated with monoallelic heterozygous variants, are associated with increased susceptibility to adult-onset neurodegeneration.

### 5.2. A PITRM1 Mutant Dog

A novel early-onset *PITRM1*-related neurodegenerative syndrome was recently described in Parson Russell Terrier dogs [71]. The disease started with epilepsy at 6–12 weeks of age, presenting seizures with orofacial automatism and twitching, repeated jerking head movements, rhythmic blinking, swallowing, salivation, and anxiety-related behaviour that progressed rapidly to *status epilepticus* and death or euthanasia. Neuropathological analysis revealed a severe acute neuronal degeneration with diffuse necrosis affecting the grey matter throughout the brain, with extensive intraneuronal mitochondrial crowding and accumulation of Aβ.

Genetic analysis identified an in-frame homozygous 6-bp deletion in *PITRM1* resulting in the loss of two amino acid residues in the N-terminal part of the protein (predicted as p.L59_S60del or, alternatively, p.S60_L61del). Bioinformatics analysis predicted that the deletion affected a region where the three-dimensional protein structure forms a short helix followed by a β-strand and that loss of the amino acids would be deleterious. Immunohistochemistry in different tissues revealed similar levels of PITRM1 protein, suggesting that structural changes potentially affected the catalytic activity necessary for precursor processing and degradation. Modelling the change in yeast confirmed the deleterious nature of the mutation, showing a 20% reduction in the catalytic activity. In contrast to the mouse model, heterozygous dogs were normal.

Lastly, a separate study on dogs affected by progressive retinal atrophy (PRA), the equivalent of retinitis pigmentosa (RP) in humans, revealed that nonsynonymous coding SNPs in *PITRM1* (p.S715A) and *APP* (p.T266M) were significantly associated with this type of neurodegeneration [72].

## 6. Cellular Models of PITRM1 Impairment

Fibroblasts derived from a Norwegian patient carrying the *PITRM1^R183Q^* mutation were the first in vitro model used to characterize cellular pathophysiology associated with PITRM1 dysfunction. Evidence of mitochondrial dysfunction in this cell line includes a significant growth defect observed on respiration-obligatory galactose medium, but not on glycolytic-permissive glucose medium, compared to *PITRM1^wt^* cells. A similar growth defect was observed in control immortalised fibroblasts stably expressing a *PITRM1*-specific shRNA, which decreased PITRM1 protein levels to approximately 40% of the amount found in cells transfected with the empty vector. All these cell lines also showed a significant reduction in the mitochondrial membrane potential and a reduced disposal of a synthetic, exogenous fluorescent Aβ_1–40_ peptide. The same findings have been observed in the *PITRM1^T931M^* mutant fibroblasts (data not published).

To study changes in mitochondrial physiology due to loss of PITRM1, a *PITRM1* knockout HEK293T cell line (PreP^−/−^) was generated [37]. Loss of PITRM1 resulted in severe mitochondrial dysfunction characterized by defects in respiratory chain complexes and membrane potential. Complex III, complex IV, and respiratory chain supercomplexes (SC) were reduced in *PITRM1*^−/−^ mitochondria compared to wild-type, while complex II was not altered. Basal and maximal respiration were significantly lower in *PITRM1*^−/−^ compared to control and the mitochondrial membrane potential decreased, compromising the mitochondrial capacity to import newly synthesized precursor proteins from the cytosol. Further, cells lacking PITRM1 displayed changes in the nuclear expression of genes associated with mitochondrial stress responses (*ATF4*, *CHAC1*, *ASNS*, and *PCK2*), which activates retrograde signaling used by the cells to respond to mitochondrial dysfunction [73]. These are referred as mitochondrial stress responses and preserve cell viability by modulating metabolic and mitochondrial pathways [74].

Mechanistic analysis of *PITRM1*^−/−^ cells demonstrated impaired MTS degradation. As a consequence, compromised presequence cleavage by MPP occurred, with accumulation of non-processed precursor proteins, revealing that presequence degradation is coupled with precursor processing that is similar to what was described in the *Cym1* (orthologue of *PITRM1*) null yeast model [75]. Furthermore, the activity of MIP (a peptidase that cleaves eight residues from a subset of precursors) was strongly impaired in import assays on isolated mitochondria, suggesting that PITRM1 also degrades cleaved octapeptides and that their accumulation in the absence of PITRM1 induces feedback inhibition of MIP.

To overcome the limitation of the embryonic lethality previously observed in *PITRM1*-knockout mice and examine the mechanistic link between PITRM1 deficiency and neurodegeneration in a model resembling human disease, *PITRM1*-knockout (*PITRM1*^−/−^) human induced pluripotent stem cells (iPSCs) were generated using CRISPR/Cas9 endonuclease [76]. This system introduced a frameshift deletion between exon 3 and exon 4, which resulted in the complete absence of the protein.

The first functional evidence of mitochondrial impairment was provided by the decreased ratio of processed, mature to immature frataxin detected in *PITRM1*^−/−^ iPSC-derived neural precursor cells and neurons, confirming that the loss of PITRM1 leads to impaired function of MPP and defects in mitochondrial presequence processing, such as that of frataxin. Notably, a similar dysregulation of frataxin maturation was found in HEK293T cells, both in the *PITRM1^−/−^* and in the *PITRM1^R183Q^* models [37]. However, whether frataxin processing and maturation are dysregulated in patients with mutated *PITRM1* has not yet been clarified.

Loss of the mitochondrial membrane potential was more evident in the neurites than in the soma in *PITRM1*^−/−^ iPSC-derived neurons compared with isogenic *PITRM1*^+/+^ control.

Accumulation of damaged or unfolded proteins in mitochondria triggers a compensatory mechanism termed mitochondrial unfolded protein response (UPRmt) [73,77,78]. UPRmt gene activation was observed in both sporadic and familial AD [77,79]. PITRM1 deficiency also strongly induced UPRmt, and iPSC-derived neurons exhibited a significant upregulation of *ATF4, DDIT3, HSP60, HSPA9, ERO1* transcripts as well as increased expression levels of the mitochondrial proteases LONP1 and CLP. This strong upregulation of the integrated stress response (ISR) pathway [80], in turn, induced autophagy activation and increased mitochondrial clearance. Furthermore, *PITRM1*-knockout neurons showed higher levels of amyloid precursor protein and Aβ, albeit without an increase in cell death. Sub-fractionation methods coupled with a highly sensitive immunoassay failed, however, to detect Aβ peptides in mitochondrial extracts from iPSC-derived nerve cells, raising the question of whether these structures enter mitochondria, at least under these experimental conditions. This is in contrast with results previously reported by several groups, documenting the presence of Aβ within mitochondria [26,81,82].

Accumulation of APP and Aβ_1–40_ peptide in cell cultures may indeed be attributed to protein backlogging in the cytoplasm caused by the accumulation of non-digested MTS, producing a general impairment of mitochondrial protein translocation and cell proteostasis. This mechanism could indeed explain the accumulation not only of APP and Aβ_1–40_ peptides but also of other proteins including, for instance, TDP-43, FUS, and Tau. That this could be driven by defective PITRM1 is supported by our preliminary data showing a skewed distribution of a potentially deleterious heterozygous *PITRM1* variant in a cohort of approximately 600 subjects with either AD or FTD and FTD-related clinical variants, compared to age-matched controls (unpublished data).

Another interesting finding in PITRM1^−/−^ neurons is the significant decrease in mitochondrial membrane potential that is more evident in neurites than in soma [74]. Assuming that this phenomenon is due to Aβ accumulation, it would support previous observations documenting a more evident aggregation of Aβ in synaptic mitochondria. However, PITRM1^−/−^ neurons did not show an increase in mitochondrial ROS production or defects in oxidative phosphorylation, which have been considered direct deleterious effects of Aβ accumulation [76].

Importantly, cerebral organoids derived from *PITRM1*-deficient iPSCs did spontaneously develop AD pathological features including protein aggregates, tau pathology, and neuronal cell death. All of these features were exacerbated by treatment with ISRIB, a global integrated stress response (ISR) inhibitor [83,84]. ISRIB-treated *PITRM1*^+/+^ cerebral organoids showed a significant increase in cleaved caspase-3-positive cells. Interestingly, ISRIB-treated cerebral organoids also showed an increase in mitochondrial DNA, suggesting that the inhibition of UPRmt leads to decreased mitochondrial clearance.

All the findings reviewed above suggest that *PITRM1* deficiency induces an impairment of mitochondrial proteostasis and the activation of UPRmt, which, in turn, activates cytosolic quality control pathways such as the ubiquitin–proteasome system (UPS) and autophagy [85]. The overload of the UPS reduces the capacity to degrade cytosolic proteins, leading to APP accumulation, the rise of Aβ species, an increased Aβ42/40 ratio, and extracellular protein aggregation. Notably, boosting mitophagy via NAD^+^ precursors partly prevented Aβ proteotoxicity [86].

The effects of PITRM1 loss may also affect immune competence. While single-cell RNA sequencing showed impaired mitochondrial function in all cell types in *PITRM1*-knockout cerebral organoids, inflammatory pathways (including iNOS, PPAR signalling TNFR1, RAR activation, chemokine, and IL-17A signalling pathways) were significantly dysregulated in astrocytes. Furthermore, *PITRM1^−/−^* cerebral organoids showed a substantial increase in the inflammatory cytokine TNF-α, which is consistent with a role of PITRM1 deficiency in immune pathways. Notably, increases in pro-inflammatory cytokines and the activation of adaptive immune response have been described in various neurodegenerative diseases [87]. For instance, AD patients showed altered plasma and CSF levels of pro-inflammatory IL-1β, IL-6, and TNF-α, as well as anti-inflammatory cytokines, IL-1 receptor antagonist, and IL-10 [88].

## 7. Conclusions

Mitochondrial protein homeostasis (proteostasis) is key for the maintenance of energetic efficiency and for protein quality control, and its impairment has been associated with human disease and neurodegeneration [89]. In this review, we collected substantial evidence suggesting that the mitochondrial peptidase PITRM1 plays an important role in cellular proteostasis. By processing the MTS of proteins transported into the inner compartment of mitochondria, PITRM1 prevents the accumulation of potentially toxic structures acting as detergents and, therefore, damaging the mitochondrial membranes. In addition, the role of PITRM1 is at the end of a chain of events, which include the cleavage of the MTS by the inner membrane peptidase, after the translocation of the precursor proteins by the TIM23 system, which acts in an energy-dependent fashion due to the exploitation of the mitochondrial membrane potential and the presence of ATP. Therefore, the block of PITRM1 digestion can determine the backlogging of proteins during their translocation into the inner compartment of mitochondria, and potentially the impairment not only of this crucial mechanism for mitochondrial homeostasis, but also the accumulation of mitochondrial protein precursors in the cytosol, eventually determining a general derangement of proteostasis in the cell. Although several reports indicate that APP and its abnormal digestion product Aβ_1–42_ can accumulate within mitochondria, where it is disposed by PITRM1, the question is still controversial. Nevertheless, investigation in the recessive human *PITRM1*-related disease as well as indifferent experimental models in vivo and in vitro show that impaired PITRM1 enzymatic activity can cause Aβ accumulation, whether in mitochondria or in the cytosol, and damage to multiple mitochondrial pathways, eventually leading to MTS and misfolded protein accumulation, increased UPRmt, and possibly dysregulation of the inflammatory response. Impaired mitochondrial proteostasis could also trigger a cytosolic response with overload and saturation of the proteasome and defective cytosolic protein degradation, eventually increasing extracellular protein aggregation (Figure 2).

Future research directions will include the assessment of diagnostic markers (e.g., monitoring PITRM1 levels or measuring the concentration of precursor proteins in patients’ blood cells) and the development of therapeutic approaches aimed to stimulate/preserve the presequence processing machinery. For the latter purpose, a possible strategy is based on the identification of new effective PITRM1 agonists through high-throughput screening or drug repurposing. An alternative option relies on the stimulation of IDE to compensate the reduced MTS degradation due to the PITRM1 deficit. Recent reports showed that the *IDE* promoter is a direct target of the Peroxisome proliferator-activated receptor gamma (PPARγ) [90]; furthermore, inositol phosphates was shown to interact with the inner chamber wall of IDE, leading to a remarkable increase in the enzyme activity [91]. Finally, approaches aimed at stimulating mitochondrial biogenesis [92] could be used in cells/patients with partly deficient PITRM1 or haploinsufficiency, since they are expected to promote some improvement by increasing the number of functional units, and possibly of the PITRM1 activity. Moreover, we showed that enhancing UPRmt and mitophagy ameliorates neuropathological features in defective *PITRM1* cells [76].

Although further studies are required to fully elucidate the pathogenic mechanism triggered by PITRM1, available data provide substantial support for its primary role in diverse mitochondrial dysfunctions associated with neurodegeneration.

## Figures and Tables

**Figure 1 biomedicines-09-00833-f001:**
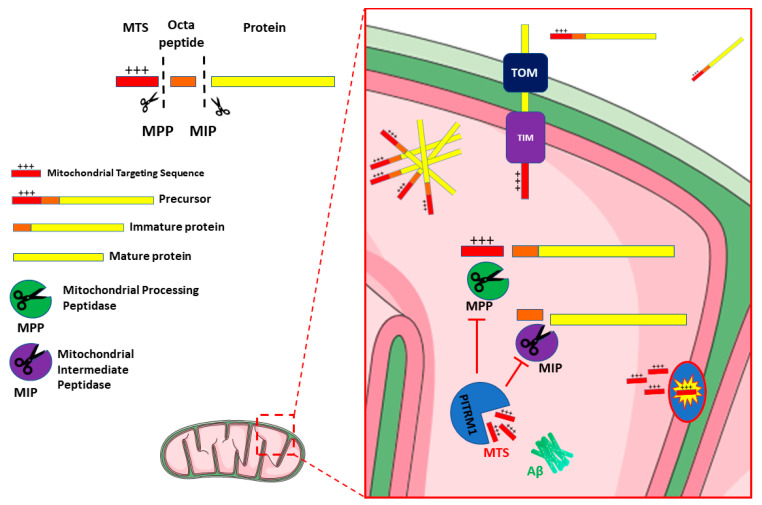
Schematic representation of PITRM1 function and interaction. Mitochondrial Precursor proteins are imported from cytosol to the matrix through the Translocase of the Outer Membrane (TOM) and Translocase of the Inner Membrane (TIM) complexes. Dysfunctional activity of PITRM1 results in the accumulation of Amyloid beta (Aβ), of Mitochondrial Targeting Sequences (MTS) and of octapeptides that trigger feedback inhibition of Mitochondrial Processing Peptidase (MPP) and Mitochondrial Intermediate Peptidase (MIP), leading to the accumulation and aggregation of unprocessed precursor. Figure 1 was modified from SMART (Servier Medical Art), licensed under a Creative Common Attribution 3.0 Generic License. http://smart.servier.com/.

**Figure 2 biomedicines-09-00833-f002:**
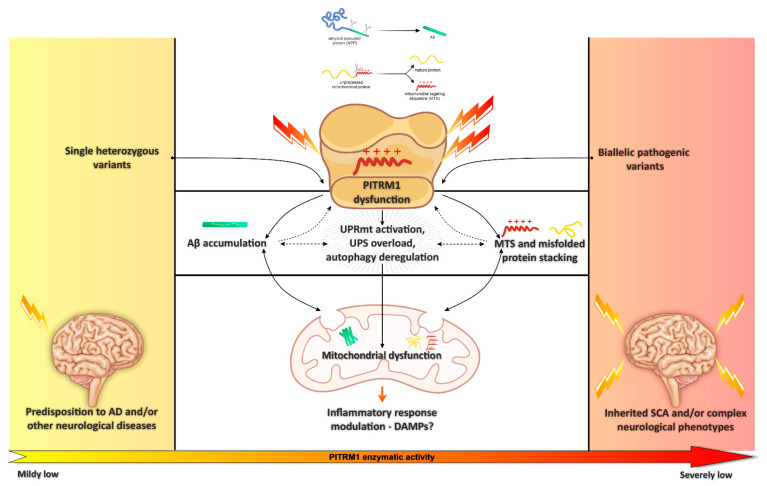
Schematic representation of the mitochondrial pathways associated with impaired PITRM1 enzymatic activity, mainly linked to Amyloid beta (Aβ) accumulation and mitochondrial targeting sequence (MTS) and misfolded protein stacking. Figure 2 was modified from SMART (Servier Medical Art), licensed under a Creative Common Attribution 3.0 Generic License. http://smart.servier.com/.

**Table 1 biomedicines-09-00833-t001:** Summary of clinical findings in patients harboring biallelic mutations in *PITRM1*.

	Patient 1 [65]	Patient 2 [65]	Patient 3 [69]	Patient 4 [69]	Patient 5 [69]	Patient 6 [69]	Patient 7 [70]
***PITRM1*** **variant**	c.548G > A	c.548G > A	c.2795C > T	c.2795C > T	c.2795C > T	c.2795C > T	c.2239dupG
**Effect on PITRM1 expression**	Markedly reduced	Markedly reduced	Markedly reduced	Markedly reduced	Markedly reduced	Markedly reduced	Markedly reduced
**Ethnicity**	Norwegian	Norwegian	Palestinian Arab	Palestinian Arab	Palestinian Arab	Palestinian Arab	Unknown
**Onset of disease**	Childhood	Childhood	5 months	Early childhood	2 years	Early childhood	25 months
**Symptoms at onset**	Mild intellectual disability	Mental retardation	Developmental delay, tremor	Developmental delay	Mild motor and cognitive impairment, dysmetria	Mild motor and speech delay	Seizures, developmental delay, hypotonia
**Main clinical findings**	Ataxic syndrome, cognitive impairment, psychosis	Ataxic syndrome, cognitive impairment, psychosis	Intellectual disability, severe ataxic syndrome, psychosis	Moderate intellectual disability, severe ataxic syndrome, psychosis	Mild developmental delay and ataxic syndrome	Mild developmental delay	Ataxic syndrome, intellectual disability
**Brain Imaging**	Marked cerebellar atrophy, unilateral thalamic signal changes	Cerebellar and mild cerebral atrophy	Cerebellar atrophy	Severe cerebellar atrophy, mild cerebral atrophy	Normal	Normal	Moderate cerebellar atrophy, bilateral thalamic signal changes
**Muscle biopsy**	COX-negative fibers	Unknown	Normal	Unknown	Unknown	Unknown	Slight mitochondrial proliferation
**OXPHOS efficiency on patients’ fibroblasts**	Unknown	Unknown	Unknown	Unknown	Unknown	Unknown	Defective
**Other findings**	Low Aβ_1–42_ CSF levels	n.r.	Increased levels of CPK, lactate and pyruvate on blood tests	No sign of muscle damage on blood tests	No sign of muscle damage on blood tests	No sign of muscle damage on blood tests	n.r.

Aβ_1–42_: amyloid beta 1–42; COX: cytochrome c oxidase; CPK: creatine phosphokinase; CSF: Cerebrospinal fluid; n.r.: not reported.

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
