# Peer review of "Role of PITRM1 in Mitochondrial Dysfunction and Neurodegeneration"

_biomedicines, 2021, doi:10.3390/biomedicines9070833_

Round 1
Reviewer 1 Report
I have no comments on the proposed review, and this review can be published.
Author Response
Since there were no comments, we have no answers
Reviewer 2 Report
This is a very interesting review which links a dysfunctional metalloprotease PITRM1 with neurodegenerative disorders with amyloid accumulation. The paper is well written and sound and clear. I have only two minor comments:
- An obvious problem for the proposed wide action of PITRM1 is the localization of PITRM1 in the mitochondrial matrix which should lead exclusively to a massive intramitochondrial Aß accumulation. Is the mitochondrial Aß exported to the cytosol ? The authors should comment on this issue in greater detail.
- Is there any evidence for the upregulation of genes involved in mitochondrial stress responses in AD patients or AD models? The proposed impaired processing of imported proteins should increase the level of free presequence peptides leading to stress responses detectable in transcriptome studies.
Author Response
This is a very interesting review which links a dysfunctional metalloprotease PITRM1 with neurodegenerative disorders with amyloid accumulation. The paper is well written and sound and clear. I have only two minor comments:
- An obvious problem for the proposed wide action of PITRM1 is the localization of PITRM1 in the mitochondrial matrix which should lead exclusively to a massive intramitochondrial Aß accumulation. Is the mitochondrial Aß exported to the cytosol ? The authors should comment on this issue in greater detail.
Aß has been reported to be present in mitochondria by several authors, and we have shown the accumulation of beta in the neuropil of PITRM1 KO mice. Alternatively, as hypothesised by Deleidi et al (and reported in the review), the absence of PITRM1 could cause a general backlogging of proteins not only in mitochondria but also in other compartments of the cell including the cytosol. Deleidi et al. do show in their iPSC-derived neurones accumulation of the Abeta precursor, not in mitochondria but in the cytosol. This general impairment of cell proteostasis could eventually lead to accumulation of Aß through an indirect mechanism. This concept is clearly expressed in the review.
- Is there any evidence for the upregulation of genes involved in mitochondrial stress responses in AD patients or AD models? The proposed impaired processing of imported proteins should increase the level of free presequence peptides leading to stress responses detectable in transcriptome studies.
There is a mounting evidence of the occurrence of mitochondrial stress and stress response in AD, for instance refer to:
Weidling I, Swerdlow RH. Mitochondrial Dysfunction and Stress Responses in Alzheimer's Disease. Biology (Basel). 2019 May 11;8(2):39. doi: 10.3390/biology8020039. PMID: 31083585; PMCID: PMC6627276.
Sharma VK, Singh TG, Mehta V. Stressed mitochondria: A target to intrude alzheimer's disease. Mitochondrion. 2021 Apr 8;59:48-57. doi: 10.1016/j.mito.2021.04.004. Epub ahead of print. PMID: 33839319
John S. Beck, Elliott J. Mufson and Scott E. Counts, “Evidence for Mitochondrial UPR Gene Activation in Familial and Sporadic Alzheimer’s Disease”, Current Alzheimer Research 2016; 13(6). https://doi.org/10.2174/1567205013666151221145445
Melber, A., Haynes, C. UPRmt regulation and output: a stress response mediated by mitochondrial-nuclear communication. Cell Res 28, 281–295 (2018). https://doi.org/10.1038/cr.2018.16
Navarro JF, Croteau DL, Jurek A, et al. Spatial Transcriptomics Reveals Genes Associated with Dysregulated Mitochondrial Functions and Stress Signaling in Alzheimer Disease. iScience. 2020;23(10):101556. Published 2020 Sep 15. doi:10.1016/j.isci.2020.101556
These studies indicate that a specialized mitochondrial unfolded protein response (UPRmt) is activated upon the aberrant accumulation of damaged or unfolded proteins in the mitochondrial matrix, resulting in the up-regulation of key genes involved in mitochondrial stabilization.
We included some sentences and additional citations on this issue in the manuscript.

Reviewer 3 Report
24 June 2021
Review on the manuscript titled “Role of PITRM1 in mitochondrial dysfunction and neurodegeneration” by Brunetti D et al., submitted to Biomedicines
Manuscript ID: biomedicines-1271242
Dear Authors,
Neurodegenerative diseases are linked to mitochondrial dysfunction leading synaptic dysfunction, apoptosis, and neurodegeneration. Degraded and misfolded mitochondrial peptides are observed to play a role in the pathogenesis. The pitrilysin metallopeptidase 1 (PITRM1) is a member of the protease-mediated quality control system. The authors review PITRM1 mutations and cellular and animal models of PITRM1 deficiency, proposing PITRM1 as an important factor for neurodegenerative diseases.
- Page 1, Abstract: Please clarify the manuscript is a review article.
- Page 1, Abstract: Please define Aβ1-42 fist and then use the abbreviation.
- Page 2, Paragraph 2: The etiology of neurodegenerative diseases was reviewed recently including mitochondrial dysfunction. Suggested reference: https://doi.org/10.3390/ijms21072431; https://doi.org/10.3390/molecules25030564
- Page 3, Figure 1: Please define the abbreviations.
- Pages 1-4, Introduction: Please restructure the Introduction. The following Section 2 should be about PITRM1.
- Pages 4,5 The Section 2: Please restructure the section with subsections.
- Pages 5,6, The Section 3: Please add a table or figure summarizing the section.
- Pages 8,10, The Section: Please add a table or figure summarizing the section.
- Page 10, Conclusion: Please summarize the key findings, weaknesses in the research, potentials, the ultimate goal, research or knowledge needed to achieve, the biggest challenge in this goal, and future research directions, among others.
- Pages 11-15, References: Please cite more references, preferably at least more than 150 for review articles.
The manuscript contains two figures, no table and 70 references. The reviewer recommends restructuring the contents of the manuscript using subsections and tables summarizing the sections. A graphic abstract may also help. The manuscript carries important value regarding the protease-mediated quality control system PITRM1. I recommend this manuscript for publication after major revision.
I declare no conflict of interest regarding this manuscript.
Best regards,
Author Response
Neurodegenerative diseases are linked to mitochondrial dysfunction leading synaptic dysfunction, apoptosis, and neurodegeneration. Degraded and misfolded mitochondrial peptides are observed to play a role in the pathogenesis. The pitrilysin metallopeptidase 1 (PITRM1) is a member of the protease-mediated quality control system. The authors review PITRM1 mutations and cellular and animal models of PITRM1 deficiency, proposing PITRM1 as an important factor for neurodegenerative diseases.
- Page 1, Abstract: Please clarify the manuscript is a review article. Editorial amendment.
This is already clear, since we wrote: “in this review, we present…” (line 28)
- Page 1, Abstract: Please define Aβ1-42 fist and then use the abbreviation. Editorial amendment.
Done: we defined it in the abstract.
- Page 2, Paragraph 2: The etiology of neurodegenerative diseases was reviewed recently including mitochondrial dysfunction. Suggested reference: https://doi.org/10.3390/ijms21072431; https://doi.org/10.3390/molecules25030564. Editorial amendment.
We added the first reference in the manuscript, in a sentence that better clarifies the link between neurodegeneration and inflammation/cytokines: the proposed references were not added in the introduction because they are poorly or not at all related to mitochondrial dysfunction.
- Page 3, Figure 1: Please define the abbreviations. Editorial amendment.
Done.
- Pages 1-4, Introduction: Please restructure the Introduction. The following Section 2 should be about PITRM1. Editorial amendment.
We added a section about PITRM1.
- Pages 4,5 The Section 2: Please restructure the section with subsections. Editorial amendment.
Done: the new section 3 is now divided in 2 subsections.
- Pages 5,6, The Section 3: Please add a table or figure summarizing the section. Editorial amendment.
According to the reviewer’s suggestion, we added a table reporting genetic and clinical features of the patients harboring biallelic PITRM1 variants.
- Pages 8,10, The Section: Please add a table or figure summarizing the section. Editorial amendment.
The main concepts of the last two sections are already summarized in figure 2.
- Page 10, Conclusion: Please summarize the key findings, weaknesses in the research, potentials, the ultimate goal, research or knowledge needed to achieve, the biggest challenge in this goal, and future research directions, among others. Editorial amendment.
We expanded the conclusion paragraph adding some sentences related to the interesting topic suggested by the reviewer, including future research directions.
- Pages 11-15, References: Please cite more references, preferably at least more than 150 for review articles. Editorial amendment.
We added a series of additional references: however, we think that citations about PITRM1, which is a relatively recent and poorly characterized protein, are complete. Obviously, it is not possible to cite all the papers about neurodegeneration, Alzheimer disease and/or mitochondrial dysfunction.
The manuscript contains two figures, no table and 70 references. The reviewer recommends restructuring the contents of the manuscript using subsections and tables summarizing the sections. A graphic abstract may also help. The manuscript carries important value regarding the protease-mediated quality control system PITRM1. I recommend this manuscript for publication after major revision.
We slightly changed the structure of our manuscript according to the reviewer’s suggestion, and added several additional references.
In the revised version, we included a table reporting genetic and phenotypic features of PITRM1 patients. We think that the figure 2 already summarizes all the main aspects of the review and may act as a graphical abstract.
Round 2
Reviewer 3 Report
14 July 2021
Review on the manuscript titled “Role of PITRM1 in mitochondrial dysfunction and neurodegeneration” by Brunetti D et al., submitted to Biomedicines
Manuscript ID: biomedicines-1271242
Dear Authors,
Neurodegenerative diseases are linked to mitochondrial dysfunction leading synaptic dysfunction, apoptosis, and neurodegeneration. Degraded and misfolded mitochondrial peptides are observed to play a role in the pathogenesis. The pitrilysin metallopeptidase 1 (PITRM1) is a member of the protease-mediated quality control system. The authors review PITRM1 mutations and cellular and animal models of PITRM1 deficiency, proposing PITRM1 as an important factor for neurodegenerative diseases.
The manuscript contains two figures, one table and 92 references. The quality of the manuscript is substantially improved. The manuscript carries important value regarding the protease-mediated quality control system PITRM1. I recommend this manuscript for publication in current form.
I declare no conflict of interest regarding this manuscript.
Best regards,
Masaru Tanaka, M.D., Ph.D.